# Ethanol–Withanolides Interactions: Compound-Specific Effects on Zebrafish Larvae Locomotor Behavior and GABAA Receptor Subunit Expression

**DOI:** 10.3390/ijms262210991

**Published:** 2025-11-13

**Authors:** Kamila Czora-Poczwardowska, Radosław Kujawski, Weronika Jarczak, Emilia Cicha, Przemysław Mikołajczak, Michał Szulc

**Affiliations:** 1Department of Pharmacology, Poznan University of Medical Sciences, 60-806 Poznan, Poland; kczora@ump.edu.pl (K.C.-P.); radkuj@ump.edu.pl (R.K.); weronika.jarczak@ump.edu.pl (W.J.); przemmik@ump.edu.pl (P.M.); 2Animal Facility, Poznan University of Medical Sciences, 60-806 Poznan, Poland; emiliacicha@ump.edu.pl

**Keywords:** ethanol, withanolides, *Withania somnifera*, GABAA receptor, zebrafish larvae, locomotor behavior

## Abstract

Concurrent consumption of ethanol (EtOH) and herbal preparations containing *Withania somnifera* (WS, ashwagandha) is increasingly common, but the neurobehavioral and molecular consequences of such interactions remain poorly characterized. This study investigated how three purified withanolides—withanolide A (WITA), withanone (WIN), and withaferin A (WTFA)—modulate the effects of acute EtOH exposure in zebrafish (*Danio rerio*) larvae. Locomotor behavior was quantified under EtOH concentrations ranging from 0 to 4.0%, and the expression of four GABAA receptor subunit genes (*gabra1, gabra2, gabrd, gabrg2*) was analyzed by qPCR. EtOH alone induced a biphasic locomotor response, with stimulation at low-to-moderate doses and suppression at higher doses. WITA and WIN modulated this pattern in a dose-dependent manner, preserving or enhancing hyperactivity, while WTFA consistently potentiated locomotor suppression. mRNA profile analysis revealed subunit-specific changes, including downregulation of *gabra1* and *gabra2*, compound-dependent regulation of *gabrd*, and complex *gabrg2* responses. These results demonstrate that individual withanolides distinctly shape behavioral and molecular outcomes of EtOH exposure, suggesting specific interactions at the level of inhibitory neurotransmission. The findings provide mechanistic insight into the combined effects of WS-derived compounds and EtOH and highlight the importance of considering such interactions in both experimental and applied contexts.

## 1. Introduction

Ethanol (EtOH) is one of the most widely used [1] and socially accepted psychoactive substances. Its consumption has significant health consequences, including an increased risk of cardiovascular diseases [2,3], neurodegenerative disorders [4,5], and cancer [6], as well as notable social impacts [7]. Globally, about 2.3 billion people consume EtOH, with an average annual intake of 5.5 L of pure EtOH per person. Harmful use contributes to nearly 3 million deaths each year—over 5% of all global mortality [8]. The situation is particularly alarming in Europe, where 88% of the population has consumed EtOH at least once [9,10]. Data from the European Monitoring Centre for Drugs and Drug Addiction indicate that in this region, EtOH causes the most severe combined health and social harm among all psychoactive substances—even more than illicit drugs [11,12]. The persistently high level of EtOH consumption and its associated public health burden remain a major challenge for modern healthcare.

The mechanism of EtOH’s biological action in the human body is highly complex. Decades of research have demonstrated its most symptomatic biphasic, dose-dependent effects: low blood concentrations induce behavioral stimulation, increased activity, and euphoria, potentially reinforcing addictive mechanisms via activation of the reward system. In contrast, higher doses produce strong sedative effects by inhibiting neural transmission [13,14,15]. At elevated concentrations in the bloodstream, EtOH can cause severe consequences, including respiratory depression and even death [16].

Based on our current best knowledge, this alcohol primarily affects the central nervous system (CNS) by allosterically modulating gamma-aminobutyric acid type A (GABAA) receptors and antagonizing N-methyl-d-aspartate (NMDA) receptors [14]. By increasing the frequency of chloride channel openings, EtOH enhances the inhibitory action of gamma-aminobutyric acid (GABA), particularly in the nucleus accumbens, ventral pallidum, bed nucleus of the stria terminalis, and amygdala—producing sedation and anxiolysis [17,18]. Inhibition of neural activity in these regions also reduces threat perception, increases impulsivity, and impairs emotional control [19,20]. Moreover, GABAergic modulation in the nucleus accumbens and ventral pallidum disinhibits dopaminergic neurons in the ventral tegmental area, elevating dopamine release in the reward pathway and reinforcing euphoria and reward-seeking behavior [21].

GABAA receptors are pentameric chloride channels assembled from a pool of 19 subunits (α1–6, β1–3, γ1–3, δ, ε, θ, π, ρ1–3). The most common combinations of these subunits include the following: α1β2γ2, α2β3γ2, and α3β3γ2 [22,23]. Subunit composition determines pharmacological sensitivity, channel kinetics, and synaptic or extrasynaptic localization [24], enabling GABAA receptors to mediate both fast phasic and slow tonic inhibition throughout the CNS.

Acute EtOH exposure can induce rapid and reversible changes in GABAA receptors. These include altered subunit phosphorylation, trafficking, and surface expression, which can occur within a short time after a single dose [25]. A well-documented effect is the sensitivity of δ-containing receptors, which mediate tonic inhibition—even low concentrations of EtOH can enhance their activity in hippocampal neurons [26,27,28]. In contrast, α1- and α2-containing receptors are more commonly associated with phasic inhibition, and their acute regulation by EtOH depends on brain region, dose, and time after exposure [26,27,28].

Chronic or repeated EtOH exposure produces a distinct set of adaptive changes. Many studies in rodents have reported a downregulation of α1 and upregulation of α4 and δ subunits in cortical and striatal regions, which is consistent with a shift from synaptic to extrasynaptic inhibition [26,27,28,29]. These adaptations are thought to contribute to tolerance, withdrawal, and dependence [30,31]. However, the direction of change is not universal: increases in α1 and γ2 have also been documented, indicating region and protocol-specific effects [24]. Genetic studies further support the importance of subunit diversity: variations in GABRA2, GABRA3, and GABRA6 proteins have been linked to EtOH dependence in humans, highlighting translational relevance [27,32,33].

At the same time, EtOH acts as an antagonist of the NMDA receptor for glutamate—the main excitatory neurotransmitter in the CNS [34]. This leads to reduced calcium and sodium influx into neurons, weakening excitatory transmission [34,35]. EtOH modulates NMDA receptors mainly in brain structures involved in memory, emotion, and motor function. Moreover, EtOH influences serotonergic, dopaminergic, and cholinergic systems, further modulating behavioral effects and enhancing its addictive potential [14]. In recent decades, growing interest in healthy lifestyles and concerns about the side effects of synthetic drugs have encouraged people to use herbal products and nutraceuticals [36,37]—often, unfortunately, simultaneously with the consumption of substances such as EtOH [38,39,40]. Among the increasingly widespread unregulated preparations, one of the most popular natural products is *Withania somnifera* (WS, Ashwagandha), with withanolides as a main group of its bioactive compounds [41], attributed to its adaptogenic, neuroprotective, and anxiolytic effects [42,43,44]. Concurrent use of EtOH and over-the-counter herbal preparations such as WS raises concerns about potential interactions, both beneficial and adverse, particularly regarding their effects on the CNS [40,45]. So far, EtOH research has focused mainly on addiction and toxicity, often overlooking its possible interactions with “everyday” xenobiotics at the non-metabolic level [46,47]. Yet, plant-based products with documented neuroactivity are becoming increasingly popular, requiring a better understanding of how even low-dose EtOH used by non-dependent individuals may alter their efficacy or safety profiles [48], not only through EtOH metabolic induction [49,50] or inhibition [51], but also via receptor-level interactions. These issues are important not only for consumers who often combine various xenobiotics in daily life [37] but also for designing novel therapies or optimizing existing treatment protocols that include traditional medications or supplements, while accounting for potential EtOH use.

WS is a traditional Ayurvedic adaptogen, which has recently gained popularity in Western countries, especially as a dietary supplement. Globally, the market value of WS products in 2023 was USD 670.6 million, with an estimated annual growth of 8.3% until 2030 [52]. The mechanisms of WS extract involve multiple molecular pathways and receptors. It exhibits anti-stress activity and regulates hormonal balance by lowering cortisol levels—an effect linked to modulation of the hypothalamic–pituitary–adrenal axis [53,54] and enhanced inhibitory neurotransmission [42,55,56]. WS also has anti-inflammatory and immunomodulatory effects, as it suppresses proinflammatory cytokines like TNF-α and IL-6, and enhances NK cell and macrophage activity, boosting immune response [43,57,58]. Its antioxidant properties help neutralize free radicals and reduce oxidative stress, which contributes to its neuroprotective actions [57,59]. Moreover, WS promotes the expression of the brain-derived neurotrophic factor (BDNF), supporting neuronal protection and regeneration [60]. At the receptor level, special attention is given to WS effects on the actions of GABAA and NMDA receptors—the same targets involved in the biological mechanism underlying EtOH’s influence. WS extract can enhance GABAA receptor activity, thereby increasing inhibitory transmission, leading to anxiolytic effects and improved sleep quality [42,55,56,57,59]. WS modulation of NMDA receptors contributes to neuroprotection by reducing glutamate-induced excitotoxicity, inhibiting excessive Ca^2+^ influx, and protecting neurons from degeneration [41,61,62].

So far, studies on WS have mainly examined full plant extracts, attributing broad health benefits to them without clearly identifying which specific compounds are responsible for observed effects [42,61,63]. So far, more than 40 different withanolides have been identified in various WS parts [41,64], but only a few, particularly withaferin A (WTFA), are typically associated with major pharmacological effects [41,42,61,63,64]. The mechanisms of action of other purified withanolides, like withanolide A (WITA) and withanone (WIN), remain poorly understood, and the literature lacks clear data on their individual activities.

Currently, little is known about WS interactions with EtOH. Only a few studies have examined how WS extracts influence EtOH-related behavior, and their results do not clarify which specific compounds are responsible for the effects. A standardized WS extract was shown to reduce EtOH motivational properties [65], alleviate EtOH withdrawal symptoms [66], and suppress dopamine transmission in the mesolimbic reward pathway triggered by morphine and EtOH [67]. However, no studies to date have examined interactions between individual purified withanolides and EtOH. The one exception is a study involving docosanyl ferulate (a ferulic acid ester, not a steroidal lactone like withanolides) from WS, which inhibited EtOH and morphine-induced conditioned place preference in mice [68].

Each withanolide has a distinct structure and biological properties [41,42,61,63,64], suggesting that their CNS effects may differ. Studies on isolated withanolides can help exclude the influence of other extract components and precisely identify their specific molecular targets in the CNS.

Notably, to the best of our knowledge, there are currently no published data on WS or withanolide concentrations in environmental matrices such as surface waters or soils; thus, environmental levels are presumed to be negligible.

The aim of this study was to investigate the effects of three selected withanolides, WITA, WIN and WTFA, on behavior and the expression of genes encoding chosen subunits of GABAA receptor—α1 (*gabra1*), α2 (*gabra2*), γ2 (*gabrg2*), and δ (*gabrd*) receptors—in a larval *Danio rerio* model under EtOH co-exposure in a concentration gradient (0–4.0%).

In selecting the above-mentioned subunits, we aimed to capture the main receptor domains involved in the biphasic effects of EtOH stimulation and reward at low doses, and sedation or anxiolysis at higher doses, while δ-containing receptors are particularly sensitive to low concentrations and mediate tonic inhibition [26,27,28]. Previous studies also suggest that WS act on GABAA receptors and confer anxiolytic or neuroprotective effects, yet the role of purified withanolides remains unclear, with scarce and inconsistent data on their effects at the subunit level [69,70]. Examining the molecular “response” of these four subunits, therefore, provides, in our opinion, a good basis to determine whether WITA, WIN, and WTFA can interact with EtOH in a subunit-specific manner and shape distinct behavioral outcomes.

Larval *Danio rerio* (zebrafish) represents a practical and translational model for CNS pharmacology. Their GABAA receptor structure and function are highly conserved with mammals, enabling direct assessment of receptor-mediated mechanisms [71,72]. Recent work shows that zebrafish GABAA subunits expressed in oocytes generate GABA-evoked currents similar to vertebrate receptors [73]. Other major neurotransmitter systems, including glutamate, serotonin, and dopamine, are also functionally comparable to those in mammals [74]. Thanks to their rapid development, small size, transparency, simple drug delivery by immersion, and the possibility of conducting experiments up to 5 dpf without ethical approval, zebrafish provide an accessible model in which molecular and behavioral assays can be combined efficiently.

## 2. Results

### 2.1. Biphasic Locomotor Response to EtOH Exposure

In the first part of the experiment, the time-binned locomotor activity of larvae recorded under alternating light–dark conditions after a prior, one-hour exposure to EtOH was assessed (Figure 1). Analysis of the data across the time bins used revealed the expected pattern of light–dark transition: activity increased during periods of darkness (hyperlocomotion in the dark phase). Importantly, low to moderate EtOH concentrations (0.5–2.0%) enhanced this response in the dark phase, whereas 4.0% EtOH suppressed locomotor activity across time bins.

Next, the values presented in Figure 1 were converted into the mean cumulative swimming distance per larva, obtained by summing the time–bin data separately for the light and dark phases, and thus reflect overall locomotor activity under each illumination condition (Figure 2).

Statistical analysis of the influence of EtOH concentration gradient on swimming distance under light–dark conditions revealed significant overall effects (EtOH effect: F (4, 694) = 109.20, *p* < 0.001; condition effect: F (1, 694) = 754.06, *p* < 0.001; interaction effect: F (4, 694) = 83.091, *p* < 0.001). Further detailed analysis based on the above-mentioned between-group differences revealed that in the control groups, larvae swam a significantly longer distance in the dark compared with the light condition (*p* < 0.001) (Figure 2). Under the light condition, larvae exposed to 2.0% EtOH (*p* < 0.001) and 4.0% EtOH (*p* < 0.005) swam significantly longer distances than the control (EtOH 0%) light group. In the dark condition, larvae treated with 0.5%, 1.0% and 2.0% EtOH swam significantly longer distances than control dark (all *p* < 0.001), and larvae treated with 4.0% swam significantly shorter distances (*p* < 0.001) than control dark. Direct comparisons between illumination conditions at the same EtOH concentrations showed significantly greater distance in darkness at 0.5%, 1.0% and 2.0% EtOH (all *p* < 0.001). No significant light–dark difference was observed at 4.0% EtOH.

### 2.2. Withanolide Modulation of EtOH-Driven Locomotor Activity

#### 2.2.1. WITA Alters EtOH-Induced Behavior

In studies on the influence of EtOH and WITA gradients on swimming distance of *D. rerio* larvae in dark conditions, statistically significant overall variabilities were observed (EtOH effect: F (4, 783) = 133.65, *p* < 0.001; WITA effect: F (3, 783) = 212.48, *p* < 0.001; interaction: F (12, 783) = 26.499, *p* < 0.001). During the entire experiment, variability within the group not receiving withanolide was identical to the description for Figure 2. Further detailed analysis based on the above-mentioned between-group differences revealed a decrease in locomotor distance was observed in the control (EtOH 0%) WITA10 and WITA500 groups, and an increase was observed in the control WITA100 group, relative to the absolute control—EtOH 0% WITA0 (Figure 3). The same pattern was observed at 0.5–1.0% EtOH across the entire WITA dose gradient (all *p* < 0.001). At 2.0% EtOH, a statistically significant decrease was observed at all WITA concentrations compared to 2.0% WITA0 (all *p* < 0.001). Changes relative to 4.0% EtOH WITA0 were observed in the WITA10 group (*p* < 0.005) and were largest in the WITA100 group (*p* < 0.001).

The control group treated with WITA100 swam a significantly longer distance compared with the WITA10 control group (*p* < 0.001), whereas the WITA500 control group swam significantly shorter distances compared with the WITA100 control group (*p* < 0.001) (Figure 3). The only significant within-group differences were observed in WITA100, where larvae showed increased swimming distance at 1.0% EtOH (*p* < 0.005) and reduced distance at 4.0% EtOH (*p* < 0.001) versus EtOH 0%. A significant increase was also observed in larvae exposed to 0.5–2.0% EtOH in the WITA100 group compared with the corresponding EtOH concentrations in WITA10 (all *p* < 0.001). The same EtOH concentrations in WITA500 were significantly lower than those in WITA100 (all *p* < 0.001).

#### 2.2.2. WIN Modifies Locomotor Response Under EtOH Exposure

In studies on the influence of EtOH and WIN gradients on swimming distance of *D. rerio* larvae in dark conditions, statistically significant overall variabilities were observed (EtOH effect: F (4, 764) = 141.065, *p* < 0.001; WIN effect: F (3, 764) = 202.081, *p* < 0.001; interaction: F (12, 752) = 18.368, *p* < 0.001). Throughout the experiment, the variability within the group not receiving withanolide was identical to that described in Figure 2. Further detailed analysis showed that in darkness, in contrast to light conditions, a reduction in locomotor distance was observed relative to the WIN0 control in the WIN10 and WIN500 control (EtOH 0%) groups (Figure 4). Variability was also noted between the corresponding EtOH WIN0 groups: at 0.5% EtOH, locomotor distance decreased in WIN10 and WIN500 (*p* < 0.005 and *p* < 0.001, respectively), whereas it increased in WIN100 (*p* < 0.005). The same pattern was observed at 1.0% and 2.0% EtOH (all *p* < 0.001). At 4.0% EtOH, locomotor distance was reduced in the WIN100 (*p* < 0.001) and WIN500 groups (*p* < 0.05) compared to the corresponding WIN0 group.

The control group treated with WIN100 swam a significantly greater distance compared with the WIN10 control group (*p* < 0.001), whereas the WIN500 control group swam a significantly shorter distance compared with the WIN100 control group (*p* < 0.001) (Figure 4). In the WIN10 and WIN500 groups, a significant reduction in swimming distance compared with the control was observed at 4.0% EtOH (*p* < 0.05 and *p* < 0.001, respectively). Within the WIN100 group, larvae exposed to 0.5–2.0% EtOH swam significantly farther compared with the control (*p* < 0.001), while at 4.0% EtOH the distance was significantly reduced (*p* < 0.001). Compared with the corresponding EtOH concentrations between WIN10 and WIN100/WIN500, a significant increase was observed in WIN100 at 0.5–2.0% EtOH (all *p* < 0.001), and a decrease at 4.0% EtOH in both WIN100 and WIN500 (*p* < 0.001). Furthermore, swimming distance in WIN500 was significantly reduced compared with WIN100 at 0.5–2.0% EtOH (*p* < 0.001).

#### 2.2.3. WTFA Alters EtOH-Induced Behavior

In studies on the influence of EtOH and WTFA gradient on swimming distance of *D. rerio* larvae in dark conditions, statistically significant overall variabilities were observed (EtOH effect b: F (4, 660) = 186.020, *p* < 0.001; WTFA effect: F (3, 660) = 80.942, *p* < 0.001; interaction: F (12, 660) = 15.843, *p* < 0.001). Throughout the experiment, the variability within the group not receiving withanolide was identical to that described in Figure 2. Further detailed analysis showed that the activity profile of larvae in darkness was similar across all WTFA concentrations and was characterized by reductions at 1.0–4.0% EtOH (1.0% *p* < 0.05 for all, and 2.0–4.0% *p* < 0.001 for all) compared to WTFA0 (Figure 5). Additionally, a significant decrease in locomotor distance was observed at 0.5% EtOH in the WTFA500 group relative to the corresponding WTFA0 group (*p* < 0.05).

No statistically significant differences were observed between the WTFA control (EtOH 0%) groups or between corresponding EtOH concentrations across WTFA concentrations (Figure 5). In the WTFA10 group, a significant increase in swimming distance was observed at 0.5% EtOH (*p* < 0.001), whereas reductions were observed at 2.0% (*p* < 0.005) and 4.0% EtOH (*p* < 0.001) compared with the control. In WTFA100, a significant increase occurred at 0.5% EtOH (*p* < 0.001) and 1.0% EtOH (*p* < 0.05), while significant decreases were detected at 2.0% and 4.0% EtOH (both *p* < 0.001) versus the control. Similarly, in the WTFA500 group, a significant increase was observed at 0.5% EtOH (*p* < 0.05), and reductions were found at 2.0% and 4.0% EtOH (both *p* < 0.001).

### 2.3. Effects on GABAA Receptor Subunit Gene Expression

#### 2.3.1. *gabra1* Expression Affected by EtOH and Withanolides

In the studies on the influence of EtOH gradient and withanolides on *gabra1* gene mRNA expression in *D. rerio* larvae statistically significant overall variabilities were observed (EtOH effect: F (4, 148) = 23.939, *p* < 0.001; withanolide effect: F (3, 148) = 60.988, *p* < 0.001; interaction: F (12, 148) = 10.487, *p* < 0.001).

Post hoc analyses indicated that EtOH alone significantly (*p* < 0.001) downregulated *gabra1* expression at all tested concentrations in control larvae (Control group), with the most pronounced suppression observed at 2.0% EtOH (Figure 6). Baseline *gabra1* expression in the H_2_O (no EtOH) group was high. However, WITA, WIN, and WTFA alone (0% EtOH = H_2_O treated groups) each caused a significant reduction in *gabra1* transcript levels compared to the control group (*p* < 0.001). These findings suggest that all three WS compounds independently downregulate *gabra1* mRNA expression in zebrafish larvae.

As mentioned above, WITA alone (0% EtOH + WITA) significantly reduced the baseline expression of *gabra1* (*p* < 0.001) (Figure 6). Moreover, EtOH significantly decreased *gabra1* transcript levels in the WITA groups at concentrations 1.0% (*p* < 0.005), 2.0%, and 4.0% EtOH (both *p* < 0.001) when compared with WITA alone. At 0.5% EtOH, *gabra1* transcript level in WITA-treated larvae was significantly higher in comparison to the control group treated with 0.5% EtOH (*p* < 0.001). Such effects were not observed for the remaining EtOH concentrations in the WITA group when compared with the proper control alcohol groups (*p* > 0.05).

In contrast to the dose-dependent effect of EtOH observed in the control and WITA groups, no statistically significant differences were found between EtOH concentrations (0–4.0%) in the WIN group (Figure 6). However, when the transcript levels were compared with the corresponding EtOH concentrations in the control groups, a significant decrease was noted at a concentration of 1.0% EtOH (*p* < 0.001) and 4.0% EtOH (*p* < 0.05).

Within the WTFA group, a significant difference was detected in larvae co-exposed to 0.5% EtOH when compared with the WTFA-treated group only (*p* < 0.001), while no such differences were observed at other EtOH concentrations (Figure 6). Nevertheless, a statistically significant increase in *gabra1* WTFA mRNA levels was observed compared to their corresponding EtOH concentrations in the control group at 0.5% and 2.0% (*p* < 0.001 in both cases), and at 4.0% EtOH (*p* < 0.005).

Differences were also observed between the WS compound groups. The lowest baseline expression at 0% EtOH concentration was observed for WIN compared to other WS compounds, whereas WITA and WFTA transcripts were at the same level (Figure 6). In particular, a significant reduction in *gabra1* transcript levels was found in WIN-treated larvae at EtOH concentrations of 0% and 0.5% (both *p* < 0.001), as well as 1.0% and 4.0% (both *p* < 0.005), compared to the WITA group. Conversely, an overall increase in *gabra1* expression was seen in the WTFA group compared to WITA, with statistically significant differences at 2.0% (*p* < 0.001) and 4.0% EtOH (*p* < 0.005). At all tested EtOH concentrations, *gabra1* mRNA levels in the WTFA group were significantly higher (*p* < 0.001) than in the corresponding WIN groups.

#### 2.3.2. *gabra2* Expression Affected by EtOH and Withanolides

In the studies on the influence of EtOH gradient and withanolides on *gabra2* gene mRNA expression in *D. rerio* larvae, statistically significant overall variabilities were observed (EtOH effect: F (4, 131) = 6.753, *p* < 0.001; withanolide effect: F (3, 131) = 28.821, *p* < 0.001; interaction: F (12, 131) = 9.627, *p* < 0.001).

Post hoc analyses showed a decrease in transcript levels in the control group at all EtOH concentrations compared to untreated larvae (H_2_O), but the differences were statistically significant from 0.5% to 2.0% EtOH concentrations, and the most pronounced reduction was observed at 1.0% EtOH concentration (all *p* < 0.001) (Figure 7). Similar to *gabra1*, the highest *gabra2* mRNA expression was observed in the control (0% EtOH) group. Likewise, WITA, WIN, and WTFA alone (0% EtOH treated groups) each caused a significant reduction in *gabra2* transcript levels compared to the untreated control (H_2_O) (*p* < 0.001).

In the WITA group, EtOH administration induced a statistically significant increase in *gabra2* expression at all concentrations tested, with the highest levels observed at 0.5% and 2–4.0% EtOH (all *p* < 0.001), followed by 1.0% EtOH (*p* < 0.05) (Figure 7). Compared with the corresponding EtOH concentrations in the control group, significant increases were observed at 1.0% (*p* < 0.005) and 2.0% (*p* < 0.05).

In WIN-treated larvae, EtOH also induced higher *gabra2* mRNA expression, with statistically significant changes at 0.5% (*p* < 0.005) and 4.0% EtOH (*p* < 0.001) compared to the 0% EtOH WIN (Figure 7). However, when compared with the corresponding EtOH concentrations in the control group, a significant reduction in transcript levels was found only at 4.0% EtOH (*p* < 0.05).

In WTFA-treated larvae, no significant differences in *gabra2* expression were observed between the groups at different EtOH concentrations (Figure 7). Nevertheless, when comparing each concentration to its counterpart in the control group, there was a significant reduction in expression at 0.5% and 2.0% EtOH (*p* < 0.05), and a more pronounced reduction at 4.0% EtOH (*p* < 0.001).

Interestingly, *gabra2* transcript levels in EtOH-exposed larvae were consistently the lowest in the WTFA group compared to the other treatment groups. It was shown that baseline gene expression (0% EtOH) was not significantly different between the WS compound groups, but a clear interaction was observed between these compounds and EtOH in modulating *gabra2* expression (Figure 7). When comparing WIN with WITA, a significant reduction in expression was observed only at 4.0% EtOH (*p* < 0.05). More pronounced differences were observed between the WTFA and WITA groups, with all EtOH concentrations added to the embryo culture medium decreasing *gabra2* expression (Figure 7). When comparing WIN with WITA, a significant reduction in expression was detected only at a concentration of 4.0% EtOH (*p* < 0.05). More pronounced differences were found between the WTFA and WITA groups, where all concentrations of EtOH added to the embryo culture medium led to a decrease in *gabra2* expression (Figure 7). The strongest reduction occurred at 0.5% and 2.0% EtOH concentrations (*p* < 0.001), followed by 1.0% and 4.0% EtOH (*p* < 0.005). Comparison of WTFA and WIN further confirmed lower *gabra2* transcript levels at all EtOH concentrations in the WTFA group. The most profound decrease was observed at 4.0% EtOH (*p* < 0.001), followed by 0.5% and 2.0% (both *p* < 0.005), and 1.0% EtOH (*p* < 0.05).

#### 2.3.3. *gabrd* Expression Affected by EtOH and Withanolides

In the studies on the influence of EtOH gradient and withanolides on *gabrd* gene mRNA expression in *D. rerio* larvae, statistically significant overall variabilities were observed (EtOH effect: F (4, 135) = 10.244, *p* < 0.001; withanolide effect: F (3, 135) = 415.712, *p* < 0.001; interaction: F (12, 135) = 20.818, *p* < 0.001).

Post hoc analyses showed a statistically significant reduction in the control group over the entire range of EtOH concentrations, with the most pronounced changes observed at 0.5–1.0% EtOH (*p* < 0.001), followed by 2.0–4.0% EtOH (*p* < 0.005) (Figure 8). Differences were also observed between the WS compound groups (without EtOH), but a significant reduction in *gabrd* expression compared to the control (H_2_O) was observed only in the WIN group (*p* < 0.001).

In the WITA group, no statistically significant differences were observed either within the group across different EtOH concentrations or compared to the corresponding EtOH concentrations in the control group (Figure 8). A dose-dependent trend towards decreased *gabrd* expression was observed, indicating that the transcriptional response in the WITA group closely resembles that in the control. Similar conclusions can be drawn for larvae exposed to WIN. In contrast, larvae treated with WTFA showed the highest *gabrd* mRNA levels among all experimental groups—both in the entire range of EtOH concentrations and under baseline (without EtOH) conditions. Within the WTFA group, a statistically significant upregulation of *gabrd* was observed across the 0.5–4.0% EtOH gradient (*p* < 0.001) compared to WTFA (0% EtOH). Furthermore, when comparing each corresponding EtOH concentration between the WTFA and control groups, a strong and significant upregulation was also seen *p* < 0.001) at each concentration.

As with previously described genes, differences were observed between groups of WS compounds. Compared with the WITA group, the WIN group showed significantly reduced *gabrd* expression at 0% EtOH (*p* < 0.005) and 0.5% EtOH (*p* < 0.05) (Figure 8). In the WTFA group, transcript levels were significantly elevated at all EtOH concentrations (0.5–4.0%, *p* < 0.001) and under baseline conditions (*p* < 0.05) compared with WITA. Furthermore, the entire WTFA group showed a marked and statistically significant increase in *gabrd* expression compared with the WIN group (*p* < 0.001).

#### 2.3.4. *gabrg2* Expression Affected by EtOH and Withanolides

In the studies on the influence of EtOH gradient and withanolides on *gabrg2* gene mRNA expression in *D. rerio* larvae, statistically significant overall variabilities were observed (EtOH effect: F (4, 119) = 14.784, *p* < 0.001; withanolide effect: F (3, 119) = 240.643, *p* < 0.001; interaction: F (12, 119) = 13.311, *p* < 0.001).

Post hoc analyses showed two statistically significant differences in the control group compared to the baseline (0% EtOH): a decrease in expression at 1.0% EtOH (*p* < 0.005) and an increase at 4.0% EtOH (*p* < 0.001) (Figure 9). Furthermore, all WS compounds administered without EtOH induced significant differences in *gabrg2* expression compared to the control (0% EtOH group)—specifically, an increase in the WITA group and a decrease in the WIN and WTFA groups (all *p* < 0.001).

In the WITA group, larvae exposed to EtOH showed significant within-group differences compared to the control (0% EtOH group), with a significant marked decrease at 0.5% EtOH (*p* < 0.001) and an increase in 2.0% (*p* < 0.05) and 4.0% EtOH (*p* < 0.001) (Figure 9). Compared to the corresponding EtOH concentrations in the control group, significant differences were noted at all concentrations, with the most pronounced changes occurring at 1–4.0% EtOH (*p* < 0.001), followed by 0.5% EtOH (*p* < 0.05). The WITA group showed the highest *gabrg2* mRNA levels among all experimental groups—both in the full EtOH gradient and in baseline conditions.

In the WIN group, a statistically significant reduction in *gabrg2* mRNA expression was observed in 4.0% EtOH compared to the WIN group with 0% EtOH (*p* < 0.005) (Figure 9). Compared with the control group at corresponding EtOH concentrations, significant reductions were observed at 2.0% (*p* < 0.05) and 4.0% (*p* < 0.001).

Larvae treated with WTFA showed a significant increase in *gabrg2* transcript levels at 0.5% EtOH compared to the WTFA 0% EtOH group (*p* < 0.001) (Figure 9). However, when comparing the WTFA group to the corresponding EtOH concentrations in the control group, a significant downregulation was observed at 2.0% and 4.0% EtOH (both *p* < 0.001).

Comparison of withanolide treatment groups showed significantly lower *gabrg2* expression at all EtOH concentrations of 1.0–4.0% (*p* < 0.001) and 0%EtOH (*p* < 0.001) in WIN and WTFA compared to WITA (Figure 9). In WIN, an additional decrease was noted at 0.5% EtOH (*p* < 0.005) vs. 0.5% EtOH WITA. Statistically significant differences were observed between the WIN and WTFA groups only at 4.0% EtOH (*p* < 0.05). Comparison of withanolide-treated groups showed significantly lower *gabrg2* expression at all concentrations of EtOH 1.0–4.0% (*p* < 0.001) and non-EtOH (*p* < 0.001) in WIN and WTFA compared with WITA. In WIN, an additional decrease was noted at 0.5% EtOH (*p* < 0.005) compared with 0.5% EtOH WITA. Statistically significant differences were observed between WIN and WTFA groups only at 4.0% EtOH (*p* < 0.05).

## 3. Discussion

The present study examined the interaction between EtOH and three purified withanolides—WITA, WIN and WTFA—in larval *D. rerio*. Our results indicate that (i) EtOH induced biphasic behavioral effects, consistent with previous observations, (ii) all three withanolides modified locomotor activity in distinct ways, and (iii) both EtOH and withanolides altered the expression of GABAA receptor subunits *(gabra1, gabra2, gabrd, gabrg2*). To our knowledge, this is the first study to systematically compare isolated withanolides in the context of acute EtOH exposure at both behavioral and molecular levels.

Acute EtOH exposure in this study produced the expected biphasic locomotor response: low to moderate concentrations (0.5–2.0%) increased the distance moved, whereas a high concentration (4.0%) suppressed locomotion. This biphasic pattern has been repeatedly described in zebrafish and other species, reflecting stimulant actions of EtOH at lower doses and sedative effects at higher doses [28,75]. Our EtOH-only controls mirrored these reports: activity was elevated at 1.0–2.0% EtOH and decreased at 4.0%. Importantly, the magnitude and direction of this effect were modulated by illumination. De Esch et al. [76] and Guo et al. [77] showed that 1.0% EtOH increases locomotion mainly during dark phases, whereas 2.0% EtOH produces stimulation predominantly under light. In our study, the biphasic profile was likewise shaped by illumination: in darkness, 4.0% EtOH not only suppressed the hyperactivity induced by lower concentrations but also reduced locomotion below water controls, whereas in light, the reduction at 4.0% EtOH appeared relative to the peak hyperactivity at 1–2.0% EtOH but did not differ from baseline control activity. This indicates that the biphasic response is more pronounced in darkness, which is consistent with the results of other researchers [77,78,79,80,81]. Increased locomotor activity of larvae in darkness is also consistent with their natural behavioral response to sudden reductions in light. While in some experimental paradigms such a response can indeed be interpreted as an anxiety-related reaction [82], in the context of acute EtOH exposure, it is more likely to reflect enhanced arousal or locomotor activation driven by dopaminergic disinhibition and stimulation of the reward circuitry [21,76,77]. Notably, light—dark behavioral assays vary across studies in terms of adaptation protocols, and these differences can significantly influence EtOH sensitivity. In our study, larvae underwent a relatively short 10 min dark acclimation before testing, followed by 40 min of alternating light/dark phases. Under these conditions, we observed a robust biphasic EtOH response, with stimulation at 0.5–2.0% and suppression at 4.0%, and the dark phases consistently amplified baseline locomotion. MacPhail et al. [81] also used a 10 min dark preincubation but conducted longer recordings (60 min across three cycles), reported that locomotor activity gradually declined over successive cycles, suggesting some degree of habituation. In contrast, Guo et al. [77] employed a prolonged 50 min light acclimation and observed a blunted initial hyperactivity in response to light-to-dark transitions. This extended light adaptation likely reduced the contrast effect of the first dark phase, producing a flatter activity profile and attenuating the early stimulatory impact of 1.0% EtOH. By comparison, our shorter dark pre-exposure preserved a strong light-to-dark contrast, thereby emphasizing the hyperlocomotor response in darkness. More recently, Hillman et al. [78] suggested that 30 min of light adaptation improves behavioral stability during dark phases and enhances reproducibility across experiments. The above studies highlight those differences in acclimation—whether initiated in the dark as in our study and [81] or in the light (as in the studies by [77,78] or without acclimation [76]—systematically shape the expression of EtOH-induced locomotor effects. Our findings likely reflect a combination of EtOH’s intrinsic biphasic action and the amplification of dark-phase hyperlocomotion resulting from the brief dark pre-exposure paradigm.

Experiments examining the effect of EtOH on the activity of *D. rerio* larvae were conducted during the light–dark cycle. However, the biphasic effect of EtOH was most evident during the dark phase, where the behavioral response was more pronounced and unambiguous. Therefore, to assess the effect of withanolides on the biphasic effect of EtOH, further experiments were analyzed during the dark phase.

In addition to confirming the expected biphasic response to EtOH, we also observed that withanolides themselves modulated larval locomotion in a compound and dose-dependent manner. When combined with EtOH, these differences translated into distinct interaction patterns. WITA10 and WITA500 flattened the typical biphasic dose–response, abolishing the usual stimulation at low to moderate EtOH concentrations, whereas WITA100 preserved or accentuated this stimulatory phase: larvae exposed to WITA100 and 1.0–2.0% EtOH covered a greater distance than WITA10/500 and WITA0, while locomotion remained suppressed at 4.0%. WIN100 consistently enhanced locomotion at low to moderate EtOH concentrations, while WIN10 and WIN500 blunted the stimulatory phase; WIN500 also tended to potentiate sedation at 4.0%. WTFA produced a mixed but consistent profile: at 2.0–4.0% EtOH concentrations, locomotion was more strongly suppressed than in EtOH-only controls—the biphasic shape was preserved but shifted toward stronger inhibition at higher EtOH concentrations. In summary, WITA100 and WIN100 most often preserved or enhanced EtOH-induced hyperactivity at low to moderate doses, whereas the same compounds at 10 or 500 µg/L often flattened the biphasic response. At all concentrations, WTFA produced a consistent suppression of the locomotor profile of *D. rerio* larvae, suggesting potentiation of EtOH-induced inhibitory effects on neural activity. Overall, WITA100 and WIN100 preserved or enhanced EtOH-induced hyperactivity, while low and high doses (10/500 µg/L) tended to blunt biphasic effects. WTFA consistently suppressed locomotor activity, particularly at higher EtOH doses.

Interpreting the increase in traveled distance requires care, as the same behavioral output can reflect different underlying processes. In larval *D. rerio*, enhanced locomotion after a sudden transition to darkness has sometimes been described as a defensive, anxiety-like response [82]. However, increased movement can also result from pharmacological stimulation or disinhibition of neural circuits [21,76,77]. In our experimental conditions, dark phases already induced robust baseline activity, and the biphasic locomotor pattern observed with EtOH became more pronounced. Therefore, the additional movement observed with WITA and WIN most likely reflects enhanced arousal or dopaminergic stimulation rather than a classic anxiety-like state, although complementary endpoints (e.g., freezing, thigmotaxis, cortisol, or dopamine markers) would be required in the future to confirm this interpretation.

Thus, we observed changes in the expression of α and δ subunits of the GABA_A_ receptor following acute EtOH exposure. These subunits mediate tonic inhibition and are associated with behavioral effects such as sedation and anxiolysis [26,27,28,29]. To explore this relationship, we conducted behavioral assessments, including locomotor activity tracking and anxiety-related paradigms (light/dark conditions) in parallel with molecular analyses. Our qPCR data show a clear pattern after acute EtOH administration: transcription of the *gabra1, gabra2, and gabrd* genes was reduced, whereas *gabrg2* mRNA levels responded in a dose-dependent manner (down at 1.0% EtOH, up at 4.0% EtOH). In this broader context, our observation that α-subunit transcripts (*gabra1/gabra2*) decrease after acute EtOH exposure is consistent with several mammalian studies reporting early reductions in α-subunits following EtOH exposure [25,29].

The dose-dependent “behavior” of *gabrg2* is particularly notable. Its suppression at moderate EtOH and elevation at high EtOH concentrations suggests that larval circuits mount distinct transcriptional responses depending on the severity of the challenge. Similar context-dependent changes in γ2 (*gabrg2*) have been described in mammalian models, where strong depressant challenges can shift γ2 dynamics in ways that may stabilize synaptic inhibition [28,83].

The above-mentioned changes provide evidence of an early transcriptional response of GABAA subunit genes to a single EtOH exposure. Such rapid modulation after acute EtOH is well documented, although the direction and magnitude vary across species, brain regions, doses, and timing [25,28,29,83].

All three withanolides also influenced subunit transcription in distinct ways. At baseline (0% EtOH), each compound reduced *gabra1* and *gabra2*, but effects on *gabrd* and *gabrg2* diverged: WIN reduced *gabrd*, WITA increased *gabrg2*, and both WIN and WTFA reduced *gabrg2*. These compound-specific patterns show that purified withanolides do not act uniformly and can differentially shape GABAA transcription. When combined with EtOH, these tendencies persisted: WITA partly counteracted the EtOH-induced decreases in *gabra2* and *gabrg2*, WIN maintained broad suppression across all tested subunits, and WTFA increased *gabra1* and *gabrd* while decreasing *gabra2* and *gabrg2*. The EtOH-induced increase in *gabra1* and *gabrd* in WTFA groups suggests a shift toward stronger inhibitory signaling, though this interpretation should be treated with caution until confirmed at the protein or functional level.

Decreases in α1/α2/δ after acute EtOH may relate to short-term adaptations underlying sedation, anxiolysis, and other behavioral effects. The partial restoration of *gabra2/gabrg2* by WITA suggests buffering against EtOH suppressive effects, whereas WIN broad downregulation may reflect reduced transcriptional flexibility. WTFA mixed pattern, with increases in *gabra1/gabrd* but decreases in *gabra2/gabrg2*, indicates a more complex modulation of inhibitory balance. While these interpretations are tentative, they are consistent with literature linking α and δ subunits to EtOH behavioral effects [27,84,85].

Finally, it is important to emphasize that qPCR data represent transcriptional activity and cannot directly determine protein expression, receptor assembly, or functional inhibition. Nonetheless, given that *gabra1, gabra2, gabrd*, and *gabrg2* are all expressed and functional in zebrafish larvae [73,86], the observed transcriptional changes are likely to be biologically meaningful. They provide novel evidence that acute EtOH alters larval GABAA receptor subunit expression in ways that can be further modified by withanolides, opening clear directions for mechanistic and functional studies.

Linking behavioral outcomes with transcriptional changes provides a useful tool in interpreting our results. The observed behavioral and transcriptional profiles indicate that the direction of locomotor change corresponded with subunit-specific modulation of GABAA receptor signaling, supporting the functional link between inhibitory tone and compound-dependent behavioral outcomes. In this study, WTFA consistently reduced locomotor activity at moderate to high EtOH concentrations, while simultaneously increasing the expression of *gabra1* and *gabrd* under EtOH exposure. This molecular–behavioral correspondence is relatively straightforward: higher α1 and δ transcript levels are associated with stronger inhibitory signaling, which, in turn, is expected to blunt activity and attenuate defensive hyperlocomotion [26,27,28]. Therefore, the observed behavioral suppression by WTFA is most likely due to underlying transcriptional changes.

WIN presented an apparently opposite profile. WIN100 promoted greater locomotion at low and moderate EtOH concentrations, particularly during dark phases, while WIN, in general, reduced GABAA subunit transcript levels under EtOH. One likely interpretation is a form of disinhibition: the lower expression of α and δ subunits could weaken inhibitory control and thereby permit greater motor output when the system is activated by EtOH, as evidenced by previous findings that reducing GABAA receptor function can enhance locomotor or anxiety-like responses [28].

WITA displayed a more nuanced and mixed pattern. Behaviorally, WITA100 preserved or enhanced EtOH-induced hyperactivity; however, at the molecular level, it partially restored *gabra2* and *gabrg2* expression relative to the EtOH-only groups. This could reflect two non-exclusive mechanisms: WITA may act on additional neurotransmitter systems (e.g., glutamatergic or monoaminergic), leading to increased movement despite stabilizing GABA-related transcripts, or the partial rescue of α2 and γ2 transcripts may represent a compensatory response aimed at counteracting EtOH-induced suppression. Since α2- and γ2-containing GABAA receptors are closely linked to EtOH sensitivity and anxiety-related processes [84,87], these transcriptional effects suggest that WITA modifies the molecular environment in which EtOH exerts its behavioral effects, even if the immediate behavioral readout is heightened locomotion.

Several practical considerations temper these interpretations. First, the dose-dependent and often non-linear behavioral effects (e.g., WIN100 versus WIN10/500) indicate specific pharmacological actions of withanolides rather than nonspecific toxicity. Second, illumination and acclimation protocols are known to influence zebrafish’s dark-phase activity. In our case, the short dark pre-exposure likely amplified anxiety-like locomotion, making these phases particularly sensitive to substances that influence arousal and stress pathways [77,78,81]. Third, the molecular changes align with established subunit roles: the α and δ subunits are strongly implicated in EtOH sedative and anxiolytic actions, while the γ2 subunit contributes to synaptic organization and benzodiazepine sensitivity [25,85].

In zebrafish at 5 dpf, GABA signaling is still in the process of maturation. At this stage, both chloride transporters—the sodium–potassium–chloride cotransporter 1 (NKCC1) and the potassium–chloride cotransporter 2 (KCC2)—are present, with NKCC1 still relatively abundant and KCC2 expression gradually increasing from the early developmental period, around 2 dpf [88,89,90,91]. Due to this developmental balance, activation of GABAA receptors may produce mixed responses, either depolarizing or inhibitory, depending on the neuronal type and brain region [88,89,90,91]. This coexistence of excitatory and inhibitory GABA signaling may represent an additional factor to consider when interpreting the compound and dose-dependent behavioral patterns observed in our study, offering a broader developmental perspective on GABA-related mechanisms in this larval model.

In summary, we believe the results of this study may have important translational implications. Concomitant use of EtOH and WS-based preparations is increasingly common, yet their potential interactions remain poorly understood. Our data show that individual withanolides can significantly modulate EtOH-induced behavioral outcomes and GABAA receptor subunit expression in a compound and dose-specific manner. This suggests that even low, non-intoxicating EtOH concentrations may influence the efficacy or safety profile of WS by altering inhibitory neurotransmission. Such effects are especially relevant in populations using WS for stress reduction, anxiolysis, or cognitive support, where occasional EtOH exposure is frequent [38,39,40,45]. Subunit-specific transcriptional changes observed here also provide mechanistic insight that may inform future therapeutic strategies or guide safe combined use of WS and EtOH-containing products.

Our study has limitations. *D. rerio* larvae represent an early developmental stage; hence, receptor expression differs from that of adult zebrafish or mammals [92,93]. We also acknowledge that region-specific analyses in older zebrafish or the use of spatial methods could further refine our observations and provide more detailed insight into receptor regulation. Additionally, future studies may include immunohistochemical localization of the examined subunits to better correlate regional expression with behavioral outcomes. Since we have measured transcriptional, but not protein-level changes, as well as functional electrophysiological activities, additional studies are required to confirm the GABAA receptor activity [94]. We focused on selected subunits, though other GABAergic components likely contribute [95]. In the future, the use of GABAA receptor mutant zebrafish may help verify the subunit-specific mechanisms underlying the observed EtOH-related effects. The authors are also aware that not only is the GABAergic system associated with the effects of EtOH in *D. rerio* larvae. We believe that verification of the activity of EtOH in the experimental system used requires further studies involving other neurotransmitter systems, i.e., the glutamatergic one. Based on current knowledge, all the tested withanolides can cross the blood–brain barrier [96], but more studies in this field may also be needed to confirm this phenomenon. In summary, future studies on withanolides–EtOH interactions should combine in-depth transcriptional and epigenetic profiling with protein-level analyses, and electrophysiological experiments, extending investigations to adult zebrafish and/or rodent models.

In addition, our behavioral readout focused primarily on locomotor distance under light–dark conditions, which provides a limited view of neurobehavioral change. Complementary endpoints such as thigmotaxis, freezing, startle, or cognitive assays would further substantiate links to GABAA-mediated mechanisms and improve interpretability [92,93,94,95].

## 4. Materials and Methods

### 4.1. Reagents

E3 medium (5 mM NaCl, 0.17 mM KCl, 0.33 mM CaCl_2_·2H_2_O, 0.33 mM MgSO_4_; pH 6.5–8.5) was prepared in-house and used as a standard zebrafish embryo medium. Dimethyl sulfoxide (DMSO, 99.9%) and Invitrogen TRIzol reagent were purchased from Thermo Fisher Scientific (Waltham, MA, USA). Withanolide A (Withanolide A phyproof^®^ Reference Substance) and withanone (Withanone phyproof^®^ Reference Substance) were obtained from PhytoLab (Vestenbergsgreuth, Germany), while withaferin was obtained from Sigma-Aldrich (St. Louis, MO, USA). For the isolation of mRNA from biological material, we use chloroform (≥99.5%), EtOH (96%), and isopropanol (≥99.5%) purchased from Sigma-Aldrich (St. Louis, MO, USA). DEPC-treated water was supplied by Thermo Fisher Scientific (Waltham, MA, USA). To reverse transcribe mRNA into cDNA, we use The Reliance Select cDNA Synthesis Kit (Bio-Rad Laboratories, Hercules, CA, USA), and for qPCR—GoTaq^®^ qPCR Master Mix from Promega (Madison, WI, USA). Gene-specific primers (sense and antisense) for the selected genes were synthesized by the Sequencing Core Facility of the Institute of Biochemistry and Biophysics, Polish Academy of Sciences (Warsaw, Poland).

### 4.2. Behavioral Assays

Zebrafish larvae at 3–5 days post-fertilization (dpf), obtained from multiple breeding pairs of the AB strain, were maintained under standard laboratory conditions (temperature 28.5 °C, well aerated aquarium water, 14 h light: 10 h dark photoperiod), according to established protocols and literature recommendations [97,98]. The experiments were conducted in the Animal Facility, Poznan University of Medical Sciences. According to current legislation, experiments performed on *Danio rerio* larvae up to 5 dpf do not require approval from an animal ethics committee.

In the first step, the effects of three concentrations (10, 100, and 500 µg/L) of selected withanolides (WITA, WIN, WTFA) on larval behavior were assessed. Each compound was dissolved in E3 medium (5 mM NaCl, 0.17 mM KCl, 0.33 mM CaCl_2_·2H_2_O, 0.33 mM MgSO_4_; pH 6.5–8.5). Due to their low water solubility (stock solution: 1 mg/mL), DMSO was used to prepare the experimental solutions, which were then diluted into E3 medium according to the assigned experimental group. The final DMSO concentration was below 0.1%, a level considered safe for *D. rerio* larvae (<1%) and without detectable effects on behavior (<0.5%) [99]. Larvae with morphological abnormalities or low viability were excluded. Substances were administered for 48 h (3–5 dpf) dissolved in E3 medium. On day five, one hour before the behavioral testing, larvae were transferred to 96-well observation plates. EtOH was added to the medium in concentrations of 0%, 0.5%, 1.0%, 2.0%, or 4.0% (*v*/*v*). All combinations of withanolides and EtOH were distributed randomly on the plate. The plates were placed in the DanioVision system (Noldus, Wageningen, The Netherlands) and allowed to adapt for 10 min in darkness before recording. The system enables the recording of larval locomotor activity under controlled alternating light conditions [100,101]. For a period of 40 min (10 min in light, then 10 min in dark and repeat), distances were recorded, following protocols similar to those described by Burgess & Granato [102] and Hillman et al. [78].

Video recordings were analyzed using EthoVision XT 14 software (Noldus, Wageningen, The Netherlands), which generated raw data for statistical analysis. Each experimental condition (type of withanolide or control, EtOH concentrations) was tested in at least four independent replicates, each including at least 10–12 larvae per group (this sampling scheme corresponds to approximately 40–48 larvae per experimental condition). Throughout the entire experiment, a constant temperature regime (28.5 °C) was maintained, and stressors were minimized.

### 4.3. Molecular Analyses

RNA isolation: Total RNA was isolated from the collected tissue of zebrafish larvae using a modified Chomczynski and Sacchi method [103]. For each sample, tissue was pooled from 8 larvae. Samples were homogenized [98,104] in a commercial RNA lysis buffer using an Omni Bead Ruptor 4 (Omni International, Kennesaw, GA, USA) at speed 5 for 90 s until a uniform suspension was achieved. After centrifugation (12,000× *g*, 4 °C, 15 min), the aqueous phase was transferred to new tubes. An equal volume of isopropanol was added, and samples were incubated at −80 °C for 1 h to precipitate nucleic acids. After thawing, samples were centrifuged again (12,000 or 7500× *g*, 4 °C, 10 min). The RNA pellet was washed with 100% isopropanol and then at least twice with 70% EtOH. After drying, the pellet was dissolved in 30 µL of DEPC-treated water. RNA concentration and purity were assessed spectrophotometrically.

cDNA synthesis: First-strand cDNA was synthesized from the isolated RNA by reverse transcription (RT) using a commercial kit following the manufacturer’s instructions (Bio-Rad Laboratories, Hercules, CA, USA). RT reactions were performed in a TC 1000-G DLAB thermocycler (Dragonlab, Beijing, China) and adjusted to a final reaction volume of 20 µL with nuclease-free water.

Gene expression analysis by rt-qPCR: Gene expression was analyzed by real-time quantitative PCR with reverse transcription (RT-qPCR) [105,106,107]. Target gene-specific primers (Table 1) were designed using Primer3 version 4.1.0 and Primer-BLAST (NCBI, accessed on 15 April 2024; https://www.ncbi.nlm.nih.gov/tools/primer-blast/). Primer specificity was verified using Nucleotide BLAST (NCBI, accessed on 15 April 2024; https://blast.ncbi.nlm.nih.gov/Blast.cgi). Reactions were performed using GoTaq^®^ qPCR Master Mix (Promega, Madison, WI, USA) and a LightCycler system (Roche Diagnostics, Basel, Switzerland) with fluorescence detection. The analyzed genes included subunits of GABAA receptors (*gabra1, gabra2, gabrd, gabrg2*). The eukaryotic elongation factor 1-alpha (*ef1a*) was used as a reference gene, previously validated as stable in zebrafish larvae [108]. Each technical replicate was performed three times. Each experimental group included at least 7 individual larvae (pooled from 8 larvae) (this sampling scheme corresponds to approximately 56 larvae per experimental condition for all Molecular Analyses).

Thermal cycling conditions were optimized for each primer set and included an initial denaturation step at 95 °C for 2 min, followed by 45 cycles of amplification. Melting curve analysis was performed at the end of the run to confirm amplification specificity. The details for each primer set are provided in Table 2.

### 4.4. Statistical Analysis

Statistical analysis of behavioral and molecular (RT-qPCR) data was performed using Statistica 13 software (StatSoft, Tulsa, OK, USA). Results from biological and technical replicates within each group were averaged and are presented as mean ± SEM (standard error of the mean). The data distribution was assessed using the Shapiro–Wilk test and found to be normal. A two-way ANOVA was used for group comparisons, with a significance level of *p* ≤ 0.05. Post hoc differences between groups were evaluated using the Tukey test with the same significance threshold (*p* ≤ 0.05).

Relative gene expression was calculated using the 2^−ΔΔCt^ method. Average Ct values from technical replicates were normalized to the reference gene (ef1α) and expressed relative to the control group (0% EtOH) [105,106,107].

## 5. Conclusions

This study offers new insights into how EtOH interacts with individual withanolides in a vertebrate model. Acute exposure to EtOH in *D. rerio* larvae caused a biphasic behavioral response and affected the transcription of key GABAA receptor subunits. WITA and WIN altered these effects in a concentration-dependent manner. They often maintained or improved EtOH-induced stimulation at low-to-moderate doses, while WTFA consistently increased locomotor suppression. All three compounds affected gene expression in distinct ways, suggesting distinct molecular mechanisms underlying their actions. These findings show that even low, non-intoxicating levels of EtOH can significantly impact the activity of WS-derived compounds, changing their behavioral and molecular outcomes. Such interactions may matter for the safe and effective use of WS-based products in humans, especially since people often encounter EtOH at the same time. Future studies should examine these mechanisms in mammals and explore potential therapeutic or safety implications.

## Figures and Tables

**Figure 1 ijms-26-10991-f001:**
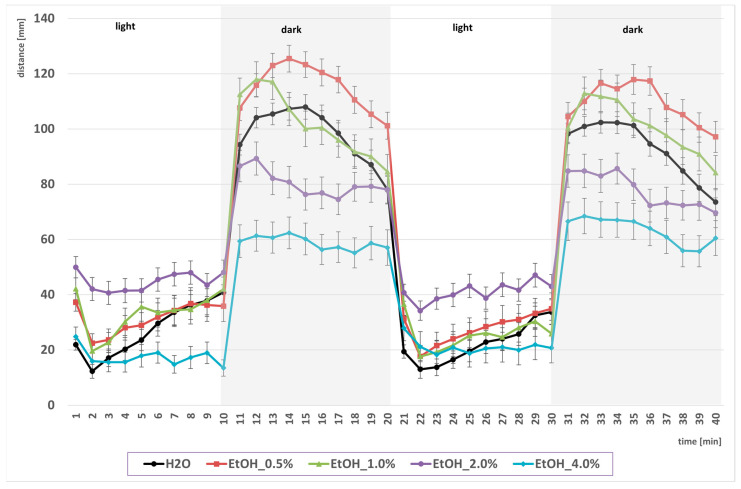
Effect of EtOH gradient on swimming distance of *D. rerio* larvae in light–dark conditions over time.

**Figure 2 ijms-26-10991-f002:**
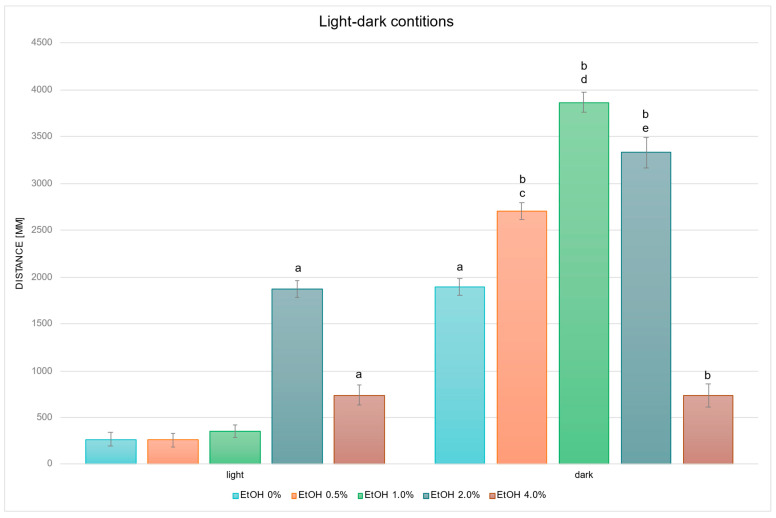
Effect of EtOH gradient on swimming distance of *D. rerio* larvae in light–dark conditions. Legend: a—vs. EtOH 0% light; b—vs. EtOH 0% dark; c—vs EtOH 0.5% light; d—vs. EtOH 1.0% light; e—vs. EtOH 2.0% light. All differences between groups were at the level: *p* < 0.05.

**Figure 3 ijms-26-10991-f003:**
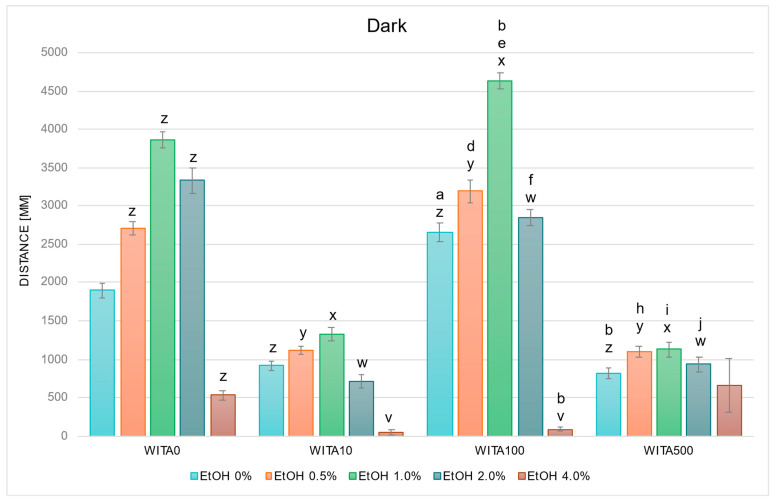
Effect of EtOH and WITA gradient on swimming distance of *D. rerio* larvae in dark conditions. Legend: z—vs. EtOH 0% WITA0; y—vs. EtOH 0.5% WITA0; x—vs. EtOH 1.0% WITA0; w—vs. EtOH 2.0% WITA0; e—vs. EtOH 1.0% WITA10; v—vs. EtOH 4.0% WITA0; a—vs. EtOH 0% WITA10; b—vs. EtOH 0% WITA100; d—vs. EtOH 0.5% WITA10; f—vs. EtOH 2.0% WITA10; h—vs. EtOH 0.5% WITA100; i—vs. EtOH 1.0%WITA100; j—vs. EtOH 2.0%WITA100. All differences between groups were at the level: *p* < 0.05.

**Figure 4 ijms-26-10991-f004:**
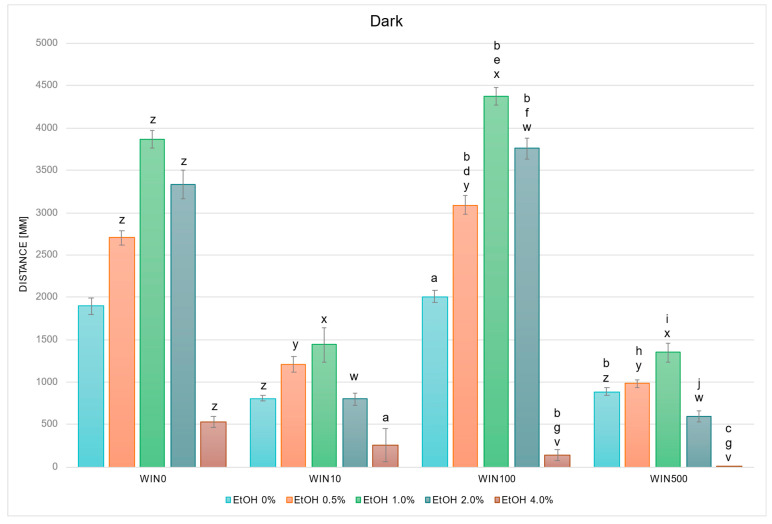
Effect of EtOH and WIN gradient on swimming distance of *D. rerio* larvae in dark conditions. Legend: z—vs. EtOH 0% WIN; y—vs. EtOH 0.5% WIN0; x—vs. EtOH 1.0% WIN0; w—vs. EtOH 2.0% WIN0; v—vs. EtOH 4.0% WIN0; a—vs. EtOH 0% WIN10; b—vs. EtOH 0% WIN100; c—vs. EtOH 0% WIN500; d—vs. EtOH 0.5% WIN10; e—vs. EtOH 1.0% WIN10; f—vs. EtOH 2.0% WIN10; g—vs. EtOH 4.0% WIN10; h—vs. EtOH 0.5% WIN100; i—vs. EtOH 1.0% WIN100; j—vs. EtOH 2.0% WIN100. All differences between groups were at the level: *p* < 0.05.

**Figure 5 ijms-26-10991-f005:**
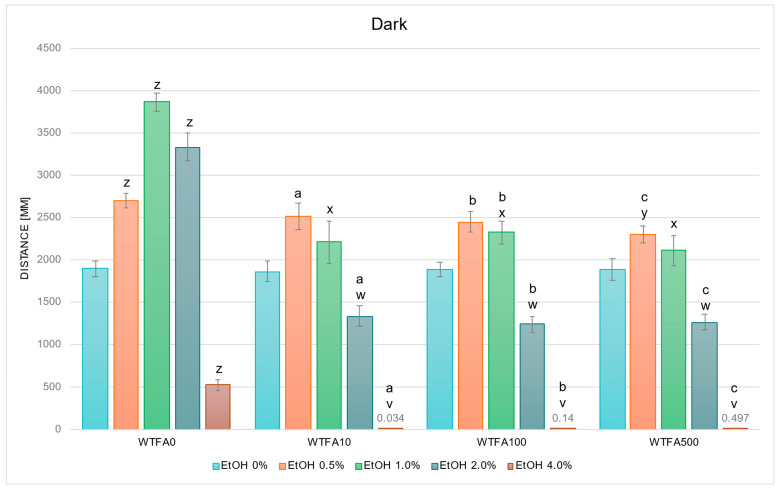
Effect of EtOH and WTFA gradient on swimming distance of *D. rerio* larvae in dark conditions. Legend: z—vs. EtOH 0% WTFA0; y—vs. EtOH 0.5% WTFA0; x—vs. EtOH 1.0% WTFA0; w—vs. EtOH 2.0% WTFA0; v—vs. EtOH 4.0% WTFA0; a—vs. EtOH 0% WTFA10; b—vs. EtOH 0% WTFA100; c—vs. EtOH 0% WTFA500. All differences between groups were at the level: *p* < 0.05.

**Figure 6 ijms-26-10991-f006:**
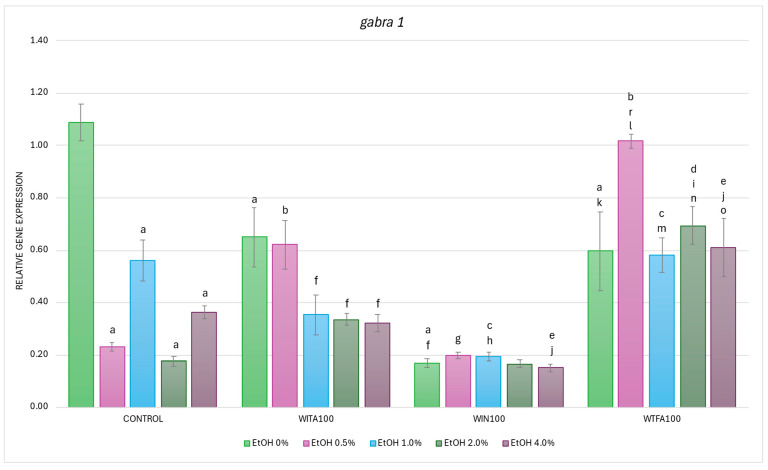
Effect of WITA, WIN and WTFA on *gabra1* mRNA expression in *D. rerio* larvae under the influence of an EtOH concentration gradient (0–4.0%). Legend: a—vs. Control (0% EtOH); b—vs. Control (0.5% EtOH); c—vs. Control (1.0% EtOH); d—vs. Control (2.0% EtOH); e—vs. Control (4.0% EtOH); f—vs. WITA (0% EtOH); g—vs. WITA (0.5% EtOH); h—vs. WITA (1.0% EtOH); i—vs. WITA (2.0% EtOH); j—vs. WITA (4.0% EtOH); k—vs. WIN (0% EtOH); l—vs. WIN (0.5% EtOH); m—vs. WIN (1.0% EtOH); n—vs. WIN (2.0% EtOH); o—vs. WIN (4.0% EtOH); r—vs. WTFA (0% EtOH). All differences between groups were at the level: *p* < 0.05.

**Figure 7 ijms-26-10991-f007:**
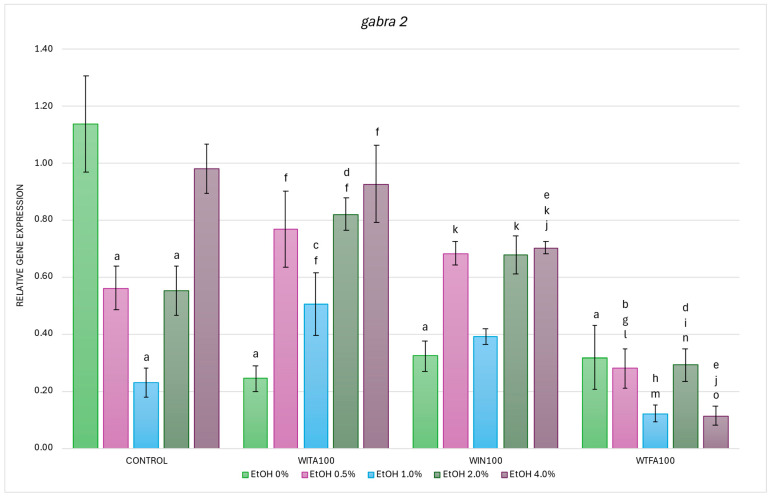
Effect of WITA, WIN and WTFA on *gabra2* mRNA expression in *D. rerio* larvae under the influence of EtOH concentration gradient (0–4.0%). Legend: a—vs. Control (0% EtOH); b—vs. Control (0.5% EtOH); c—vs. Control (1.0% EtOH); d—vs. Control (2.0% EtOH); e—vs. Control (4.0% EtOH); f—vs. WITA (0% EtOH); g—vs. WITA (0.5% EtOH); h—vs. WITA (1.0% EtOH); i—vs. WITA (2.0% EtOH); j—vs. WITA (4.0% EtOH); k—vs. WIN (0% EtOH); l—vs. WIN (0.5% EtOH); m—vs. WIN (1.0% EtOH); n—vs. WIN (2.0% EtOH); o—vs. WIN (4.0% EtOH). All differences between groups were at the level: *p* < 0.05.

**Figure 8 ijms-26-10991-f008:**
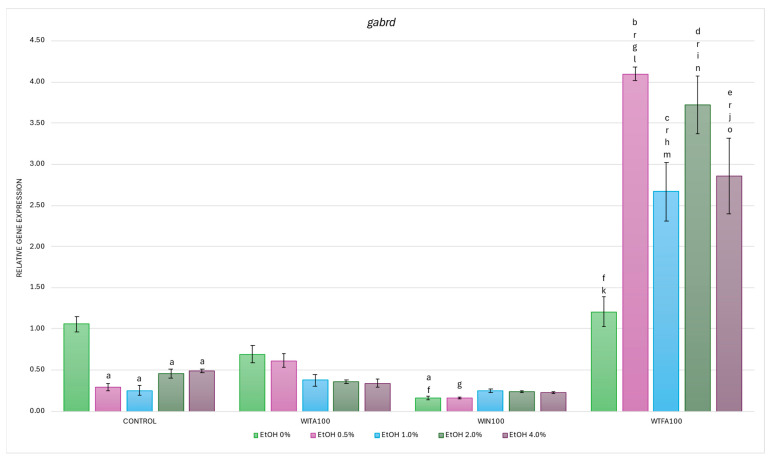
Effect of WITA, WIN and WTFA on *gabrd* mRNA expression in *D. rerio* larvae under the influence of EtOH concentration gradient (0–4.0%). Legend: a—vs. Control (0% EtOH); b—vs. Control (0.5% EtOH); c—vs. Control (1.0% EtOH); d—vs. Control (2.0% EtOH); e—vs. Control (4.0% EtOH); f—vs. WITA (0% EtOH); g—vs. WITA (0.5% EtOH); h—vs. WITA (1.0% EtOH); i—vs. WITA (2.0% EtOH); j—vs. WITA (4.0% EtOH); k—vs. WIN (0% EtOH); l—vs. WIN (0.5% EtOH); m—vs. WIN (1.0% EtOH); n—vs. WIN (2.0% EtOH); o—vs. WIN (4.0% EtOH); r—vs. WTFA (0% EtOH). All differences between groups were at the level: *p* < 0.05.

**Figure 9 ijms-26-10991-f009:**
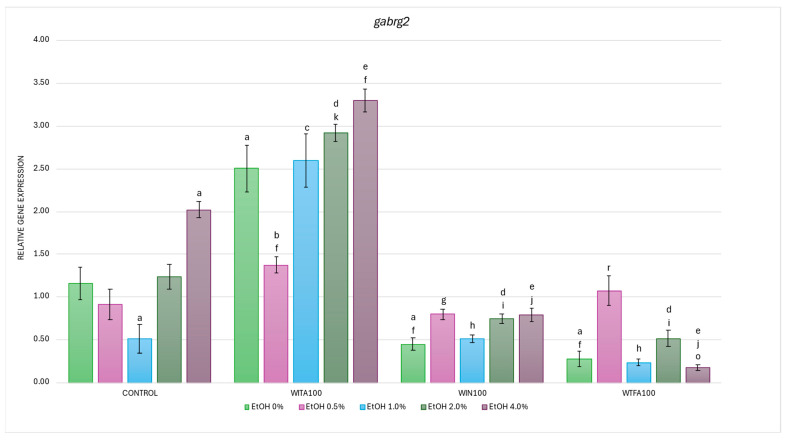
Effect of WITA, WIN and WTFA on *gabrg2* mRNA expression in *D. rerio* larvae under the influence of EtOH concentration gradient (0–4.0%). Legend: a—vs. Control (0% EtOH); b—vs. Control (0.5% EtOH); c—vs. Control (1.0% EtOH); d—vs. Control (2.0% EtOH); e—vs. Control (4.0% EtOH); f—vs. WITA (0% EtOH); g—vs. WITA (0.5% EtOH); h—vs. WITA (1.0% EtOH); i—vs. WITA (2.0% EtOH); j—vs. WITA (4.0% EtOH); k—vs. WIN (0% EtOH); o—vs. WIN (4.0% EtOH); r—vs. WTFA (0% EtOH). All differences between groups were at the level: *p* < 0.05.

**Table 1 ijms-26-10991-t001:** Primer sequences and accession numbers for reference and target genes used in RT-qPCR.

Gene	Accession Number	Forward Primer (5′-3′)	Reverse Primer (5′-3′)
*ef1a*	FJ915061.1	CTGGAGGCCAGCTCAAACAT	ATCAAGAAGAGTAGTACCGCTAGCATTAC
*gabra1*	NM_001077326.1	TGAGTCAGAGACAAGAGTGTTC	CTTCCACCCCACATCATTCTC
*gabra2*	LC596832.1	CAGACACTTTCTTTCATAACGG	TCCTCAAGATGCATTGGG
*gabrd*	XM_695007.8	AACTTTCGTCCAGGGATCGG	TGGTGTATTCCATGTTGGCTTC
*gabrg2*	NM_001256250.1	ACGGCTATGGACCTCTTCGT	TTTGAGGAAAAGAGCCGCAGG

**Table 2 ijms-26-10991-t002:** Thermal cycling conditions for the RT-qPCR amplification.

Primer	Initial Denaturation	Denaturation (45 Cycles)	Annealing	Melting Curve Analysis
*ef1a*	95 °C, 120 s	95 °C, 15 s	59 °C, 60 s	95 °C to 60 °C
*gabra1, gabra2, gabrd, gabrg2*			60 °C, 60 s	

## Data Availability

The data presented in this study are available on request from the corresponding author. The data are not publicly available due to the large volume of the raw experimental files (multiple EthoVision datasets totaling several terabytes), which makes public deposition technically and logistically impractical.

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
