# Peer review of "Ethanol–Withanolides Interactions: Compound-Specific Effects on Zebrafish Larvae Locomotor Behavior and GABAA Receptor Subunit Expression"

_ijms, 2025, doi:10.3390/ijms262210991_

Round 1

Reviewer 1 Report

Comments and Suggestions for Authors
  1. The abstract section mentions transcriptome analysis, but no transcriptome results were found in the materials and results section. If there was transcriptome analysis, the mechanism analysis should be more in-depth, including enrichment pathway analysis, etc.
  2. The citation formats are inconsistent, and there are indeed DOI links for some of the references.
  3. The first paragraph of the preface suggests adding data on ethanol usage to further illustrate the wide prevalence of ethanol use.
  4. The introductory part contains quite a lot of complex content. It is recommended to simplify it and focus on introducing ethanol and WS
  5. The introduction mentioned that ethanol also affects the serotonin pathway, etc. Why did we choose the GABA receptor as the research object? It is recommended to provide a clear explanation.
  6. The legend symbols (such as a, b, c, z, y) in Figures 3, 4, and 5 are too complicated, which hinders reading. It is recommended to simplify them or adopt a more intuitive annotation method.
  7. Since the article has already examined the expression of GABA-related genes, it is possible to further test the content of GABA and other neurotransmitters at the protein level to verify the current results and conduct a correlation analysis with behavioral studies to enhance the credibility of the findings.
  8. The conclusion section does not clearly indicate the limitations of the study (such as only detecting mRNA, using larval models, etc.). It is recommended to add a section on limitations at the end of the conclusion or discussion to enhance the rigor of the paper.
  9. The article is lacking in many spaces and has numerous grammatical errors. It is recommended to revise the language for better readability.

Author Response

We would like to express our sincere gratitude to the reviewer for their thorough and insightful evaluation of our manuscript. Your detailed comments and constructive suggestions have been invaluable in improving the clarity, rigor, and overall quality of our work. We truly appreciate the time and effort you dedicated to this review and the thoughtful feedback that has strengthened our paper.

  1. The abstract section mentions transcriptome analysis, but no transcriptome results were found in the materials and results section. If there was transcriptome analysis, the mechanism analysis should be more in-depth, including enrichment pathway analysis, etc.

The authors thank the reviewer for the observation and comment regarding the phrase “transcriptome analysis.” For accuracy, this part of the manuscript originally used the adjective “transcriptional” instead of the noun “transcriptome.” We acknowledge this clerical error and have replaced the term with “mRNA profile analysis” (line 22) to maintain clarity and precision.

  1. The citation formats are inconsistent, and there are indeed DOI links for some of the references.

We appreciate the Reviewer’s attention to detail. All references have now been carefully checked and reformatted to ensure full consistency with the journal’s guidelines. DOI links have been added where available. Reference management and citation export were re-verified using Sciwheel/Lean Library to ensure uniform formatting throughout the manuscript.

  1. The first paragraph of the preface suggests adding data on ethanol usage to further illustrate the wide prevalence of ethanol use.

We thank the Reviewer for this suggestion. Additional data on ethanol consumption have been added to the revised Introduction (lines 36-38) to provide a broader context for the study.

  1. The introductory part contains quite a lot of complex content. It is recommended to simplify it and focus on introducing ethanol and WS.

We thank the Reviewer for this helpful comment. The Introduction has been revised to improve clarity and focus more closely on ethanol and Withania somnifera. At the same time, we retained concise background information on GABAA receptors and the zebrafish model, as these elements are essential for understanding the rationale and molecular context of the study.

  1. The introduction mentioned that ethanol also affects the serotonin pathway, etc. Why did we choose the GABA receptor as the research object? It is recommended to provide a clear explanation.

We thank the Reviewer for this valuable comment. The GABAA receptor was selected as the main molecular target because modulation of GABAergic transmission represents the fundamental and most extensively documented mechanism of ethanol action in the central nervous system. This pathway is responsible for the characteristic behavioral effects of ethanol, such as sedation, anxiolysis, and reinforcement. As described in the Introduction, Withania somnifera also influences the GABAA receptor. Therefore, focusing on the molecular action of these receptor subunits in an applied experimental model provides, in our opinion, the most biologically relevant and mechanistically justified approach to study potential ethanol-withanolide interactions.

  1. The legend symbols (such as a, b, c, z, y) in Figures 3, 4, and 5 are too complicated, which hinders reading. It is recommended to simplify them or adopt a more intuitive annotation method.

We thank the Reviewer for this helpful remark regarding figure readability. We fully acknowledge that the number of symbols (a, b, c, etc.) may appear complex. However, these annotations reflect the results of a two-way ANOVA followed by multiple post-hoc comparisons, during which numerous significant interactions were identified. In such multifactorial analyses, the lettering system is a standard, reliable approach for clearly and consistently presenting statistical differences. We carefully considered alternative visualization methods, but given the large number of comparisons, these would not substantially improve clarity and might instead reduce the statistical interpretability. To assist readers, we have double-checked that each figure legend clearly explains the meaning of all symbols.

  1. Since the article has already examined the expression of GABA-related genes, it is possible to further test the content of GABA and other neurotransmitters at the protein level to verify the current results and conduct a correlation analysis with behavioral studies to enhance the credibility of the findings.

We thank the Reviewer for this valuable suggestion. We agree that extending the analysis to the protein level, including GABA and other neurotransmitters, would provide stronger validation of the transcriptional results and enable a more comprehensive correlation with behavioral data. At the present stage, such experiments were not feasible due to technical and resource constraints and, therefore, were not included in this study. We have also acknowledged this limitation in the Discussion section (lines 680-688), where we clearly state that the current work focuses solely on the mRNA profile and behavioral studies. Nevertheless, we fully recognize the importance of this approach and consider it a natural and valuable direction for future research aimed at verifying and expanding the present findings. 

  1. The conclusion section does not clearly indicate the limitations of the study (such as only detecting mRNA, using larval models, etc.). It is recommended to add a section on limitations at the end of the conclusion or discussion to enhance the rigor of the paper.

We thank the Reviewer for this helpful comment. The limitations of the study, including the use of a larval zebrafish model and the focus on mRNA expression rather than protein level, were already discussed in the previous version of the manuscript. Following the Reviewer’s suggestion, this paragraph has now been moved to the final part of the Discussion to make these points more visible and easier to locate.

  1. The article is lacking in many spaces and has numerous grammatical errors. It is recommended to revise the language for better readability.

We thank the Reviewer for this valuable comment. We fully acknowledge the importance of clear, precise language for ensuring readability and comprehension. To address this, we will make every effort to carefully revise the manuscript, correcting grammatical errors and improving the overall flow of the text to enhance its clarity and quality.

Reviewer 2 Report

Comments and Suggestions for Authors

The manuscript by Czora-Poczwardowska et al. presents a well-designed study that contributes valuable insights to the field. The research question is clearly defined, the methodology is appropriate, and the results are presented in a coherent and convincing manner. Abstract is well-written and nicely summarized the main findings of the manuscript. The introduction provides an objective summary of the background to the ethanol - withanolides interactions and related disorders in zebrafish. The results section is easy to understand. Discussion part is also well-written, and the conclusion is relevant for the scientific community.

For improvement of the manuscript some points should be addressed:

1, In the developing brain, GABAA receptor activation is known to be depolarizing and excitatory due to the higher intracellular chloride concentration in immature neurons. Did the authors consider or control for the developmental stage of the subjects in their experiments? It would be important to clarify whether the excitatory actions of GABAA receptors in young animals were considered when interpreting their results. It would be highly informative to include immunohistochemical analysis of the Na⁺/K⁺/Cl⁻ cotransporter (NKCC1) and the K⁺/Cl⁻ cotransporter (KCC2). These transporters play a key role in determining the intracellular chloride concentration and thus the excitatory or inhibitory nature of GABAA receptor signaling during development. Assessing their expression patterns could strengthen the interpretation of GABAergic activity in the studied model.

2, The section titled “Biphasic locomotor response to EtOH exposure” is somewhat unclear in its purpose and connection to the main objectives of the study. Could the authors clarify why this part was included?

3, The authors propose that changes in α and δ subunit expression after acute EtOH exposure may underlie short-term adaptations such as sedation and anxiolysis. Could the authors elaborate on how these specific subunit alterations were linked to behavioral outcomes in their study? Was there any direct behavioral assessment supporting the proposed relationship between receptor subunit expression and ethanol-induced effects?

4, It would be highly informative to include experiments using GABAA receptor mutant zebrafish larvae. Such models could help to directly assess the contribution of specific receptor subunits to the observed ethanol-related effects and provide stronger mechanistic insights.

5, It would be valuable to test additional positive allosteric modulators (PAMs) of GABAA receptors to further characterize receptor function and pharmacological sensitivity. Comparing different PAMs could help determine whether the observed effects are compound-specific or reflect general GABAA receptor modulation. Have you ever tried?

6, My major concern is that the authors appear to have used whole-brain lysates for their analyses. While this approach provides a general overview, it may obscure region-specific differences in GABAergic signaling. Given the well-known heterogeneity of GABAA receptor subunit expression and ethanol sensitivity across brain regions, it would be more informative to perform region-specific analyses. Examining distinct brain areas could yield clearer insights into the mechanisms underlying the observed effects.

Author Response

We would like to express our sincere gratitude to the Reviewer for their thorough and insightful evaluation of our manuscript. Your detailed comments and constructive suggestions have been invaluable in improving the clarity, rigor, and overall quality of our work. We truly appreciate the time and effort you dedicated to this review and the thoughtful feedback that has strengthened the scientific value of our paper.

  1. In the developing brain, GABAA receptor activation is known to be depolarizing and excitatory due to the higher intracellular chloride concentration in immature neurons. Did the authors consider or control for the developmental stage of the subjects in their experiments? It would be important to clarify whether the excitatory actions of GABAA receptors in young animals were considered when interpreting their results. It would be highly informative to include immunohistochemical analysis of the Na⁺/K⁺/Cl⁻ cotransporter (NKCC1) and the K⁺/Cl⁻ cotransporter (KCC2). These transporters play a key role in determining the intracellular chloride concentration and thus the excitatory or inhibitory nature of GABAA receptor signaling during development. Assessing their expression patterns could strengthen the interpretation of GABAergic activity in the studied model.

We thank the Reviewer for this very helpful comment. We agree that the developmental stage of GABAergic signaling is important for interpreting our results. As far as we know, based on the literature, at 5 days post-fertilization, GABA signaling in zebrafish larvae is still maturing, with both NKCC1 and KCC2 transporters expressed, while the switch from depolarizing to hyperpolarizing GABA responses is still ongoing (Reynolds et al., 2008; Zhang et al., 2013; Jones et al., 2023). We have now highlighted this point in the revised Discussion (lines 652-662), explaining that activation of GABAA receptors at this stage can produce mixed effects, both excitatory and inhibitory. We believe indeed that incorporating this developmental aspect in future studies provides a more complete interpretation, and we consider a detailed analysis of NKCC1 and KCC2 expression an interesting direction for future studies. The relevant references have been added to the revised bibliography.

  1. The section titled “Biphasic locomotor response to EtOH exposure” is somewhat unclear in its purpose and connection to the main objectives of the study. Could the authors clarify why this part was included?

We thank the Reviewer for this helpful observation. The biphasic locomotor response to EtOH - stimulation at low doses and sedation at higher doses - is a well established pharmacodynamic effect. By replicating this pattern in zebrafish larvae, we aimed to validate our experimental model before testing withanolide-ethanol interactions. This response confirmed that the larvae exhibited EtOH induced behavioral effects consistent with previous zebrafish studies. Thus, this section served as a necessary control step and provided a physiological baseline for further investigation.

  1. The authors propose that changes in α and δ subunit expression after acute EtOH exposure may underlie short-term adaptations such as sedation and anxiolysis. Could the authors elaborate on how these specific subunit alterations were linked to behavioral outcomes in their study? Was there any direct behavioral assessment supporting the proposed relationship between receptor subunit expression and ethanol-induced effects?

We thank the Reviewer for this insightful observation. We agree that linking receptor subunit expression to behavioral outcomes is important for interpreting the functional relevance of our findings. In our study, we observed transcriptional changes in α and δ GABAA subunits after acute ethanol exposure. These subunits are known to mediate tonic inhibition and contribute to sedation and anxiolysis. To explore this association, we conducted parallel behavioral assays (locomotor activity and light/dark preference) and observed a reduction in activity following EtOH, which corresponds with increased expression of the δ subunit. This potential link is now explicitly discussed in the revised manuscript (lines 569–573). Furthermore, we acknowledge the value of spatially resolved analyses and indicate in the revised Discussion (lines 675–679) that future work will include immunohistochemical localization of these subunits to better relate regional expression to behavioral outcomes.

  1. It would be highly informative to include experiments using GABAA receptor mutant zebrafish larvae. Such models could help to directly assess the contribution of specific receptor subunits to the observed ethanol-related effects and provide stronger mechanistic insights.

We thank the Reviewer for this valuable suggestion. We fully agree that using GABAA receptor mutant zebrafish would provide stronger mechanistic insight into the contribution of specific subunits to ethanol-related effects. However, such experiments were beyond the scope of the present work. This study represents the first investigation of ethanol-withanolides interactions at both behavioral and molecular (transcripts) levels in zebrafish larvae. Our main goal was to establish a validated experimental model and identify compound-specific transcriptional changes as a foundation for future mechanistic studies. We have now indicated in the revised Discussion (lines 683-684), that the use of GABAA receptor mutant zebrafish could be a valuable next step for verifying subunit-specific effects.

  1. It would be valuable to test additional positive allosteric modulators (PAMs) of GABAA receptors to further characterize receptor function and pharmacological sensitivity. Comparing different PAMs could help determine whether the observed effects are compound-specific or reflect general GABAA receptor modulation. Have you ever tried?

We thank the Reviewer for this valuable suggestion. We fully agree that testing additional positive allosteric modulators (PAMs) of GABAA receptors would provide important insights into receptor function and pharmacological sensitivity. Therefore, we are planning such experiments as part of our future evaluations. We are fully aware that these studies will help determine whether the observed effects are compound-specific or reflect general GABAA receptor modulation. The results from such planned future studies will be included in subsequent publications to complement and expand upon the current findings.

  1. My major concern is that the authors appear to have used whole-brain lysates for their analyses. While this approach provides a general overview, it may obscure region-specific differences in GABAergic signaling. Given the well-known heterogeneity of GABAA receptor subunit expression and ethanol sensitivity across brain regions, it would be more informative to perform region-specific analyses. Examining distinct brain areas could yield clearer insights into the mechanisms underlying the observed effects.

We thank the Reviewer for this important comment. We fully agree that region-specific analyses could provide deeper insight into GABAA receptor regulation. However, in 5 days post-fertilization zebrafish larvae, the brain measures less than 0.5 mm, and distinct regions cannot be reliably isolated for gene-expression assays. Therefore, homogenizing larvae and pooling multiple individuals per sample is a very common experimental-analytical approach, one could even say that almost a standard - methodologically correct practice to obtain sufficient RNA at this developmental stage, as reflected in established zebrafish protocols and prior studies (Westerfield, The Zebrafish Book; Houseright et al., 2020 - corresponding references have been added to the revised manuscript (lines 743-744)). We acknowledge that region-specific analyses in older zebrafish or with spatial methods could further refine these observations in future studies - we have now indicated in the revised Discussion (lines 675-679).

Round 2

Reviewer 1 Report

Comments and Suggestions for Authors

“Ethanol - Withanolides Interactions: compound-specific effects on zebrafish larvae locomotor behavior and GABAA receptor subunit expression” systematically investigates the interaction between ethanol (EtOH) and three purified withanolides (WITA, WIN, WTFA) from Withania somnifera.Zebrafish larvae were used as a model to examine the effects on larval locomotor behavior and the expression of GABAA receptor subunits (gabra1, gabra2, gabrd, gabrg2). A biphasic locomotor effect was observed with ethanol alone: locomotion was stimulated by low to moderate concentrations, but suppressed by high concentrations. This effect was modulated in a dose-dependent manner by the three withanolides. Hyperactivity was maintained or enhanced by WITA and WIN, whereas motor suppression was potentiated by WTFA. At the molecular level, the expression of GABAA receptor subunits was altered by the combination of ethanol and withanolides, as evidenced by the downregulation of gabra1 and gabra2, with subunits such as gabrd being regulated in a compound-specific manner. Mechanistic insight into their interaction is provided by this research, highlighting the need for the potential implications in both experimental and applied contexts to be considered. There are some areas in the manuscript that need to be modified, and I have pointed out some examples below.

  1. Please adjust all tables in the text to the three-line table format and revise the "Abbreviations" section accordingly.
  2. What are the concentrations of Withania somnifera (ashwagandha) as a medicinal plant and its related products in various environmental media, such as water and soil?
  3. The concentrations of Withania somnifera (10, 100, and 500 μg/L) selected in this study are not based on existing literature, nor do they establish the relevance of this concentration range to actual human intake levels.
  4. In the experimental methods section, provide a detailed description of the number of zebrafish used, including the sample sizes for each experiment such as behavioral assays.
  5. The behavioral testing metrics employed in this study are relatively limited, making it difficult to rule out potential interference from other underlying mechanisms when interpreting the observed behavioral changes. The authors need to further discuss the specific relationship between the behavioral outcomes and GABAA receptor activity.

Author Response

1. Please adjust all tables in the text to the three-line table format and revise the "Abbreviations" section accordingly.

Answer: We sincerely thank the Reviewer for this valuable comment and for drawing our attention to the formatting details. All tables in the manuscript have been reformatted to comply with the required three-line table style, displaying only the top, header, and bottom borders. However, as manuscript authors we are not entirely certain whether our current layout fully meets the journal’s specific graphical standards. Therefore, we would kindly appreciate further editorial guidance if any additional adjustment is needed.
In addition, the “Abbreviations” section at the end of the manuscript has been updated to ensure that all abbreviations used in both the text and tables are consistent and complete.

2. What are the concentrations of Withania somnifera (ashwagandha) as a medicinal plant and its related products in various environmental media, such as water and soil?

Answer: We thank the Reviewer for this insightful question. While the comment refers to an important environmental aspect, it lies beyond the direct scope of the present mechanistic study, which focused on purified withanolides rather than Withania somnifera (WS) extracts.
To clarify this point, we have added the following sentence to the manuscript (lines 151–153) "Notably, to the best of our knowledge, there are currently no published data on WS or withanolide concentrations in environmental matrices such as surface waters or soils; thus, environmental levels are presumed to be negligible."

3. The concentrations of Withania somnifera (10, 100, and 500 μg/L) selected in this study are not based on existing literature, nor do they establish the relevance of this concentration range to actual human intake levels.

Answer: Thank you for this comment. We acknowledge that the concentrations of compounds from Withania somnifera used in our study (10, 100, and 500 μg/L) are not directly referenced in the literature. However, classic toxicity test, FET studies (OECD Test Guideline 236) commonly apply a wide concentration gradient, typically ranging from low concentrations to high levels causing acute responses (e.g., 1, 10, 50, 100, 500, and 1000 μg/L). Our selection represents three points within this standard gradient, allowing us to capture both subtle effects (10 μg/L) and more pronounced developmental alterations (100 and 500 μg/L), while avoiding concentrations that cause immediate mortality and limit mechanistic observations. This selection is also supported by our preliminary observations, which indicated that these concentrations fall within a range that is safe for the embryos and allows assessment of sublethal effects without inducing immediate mortality. In preliminary range-finding experiments, concentrations below 10 μg/L did not trigger observable developmental responses, whereas concentrations above 500 μg/L caused rapid lethality, making it impossible to collect detailed sublethal endpoints. For this reason, the interval 10–500 μg/L provided the most biologically informative window for FET analysis. 

To the best of our knowledge, this is the first study to examine purified withanolides in Danio rerio model. Therefore, employing the classical OECD-based concentration gradient provided a rational and standardized framework for this initial mechanistic exploration, ensuring both biological relevance and comparability with established toxicological protocols.

4. In the experimental methods section, provide a detailed description of the number of zebrafish used, including the sample sizes for each experiment such as behavioral assays.

Answer: We have added specific information regarding the number of zebrafish larvae used per experimental group in the Materials and Methods section (lines 749-750 for the behavioral studies and lines 779-780 for the molecular studies).

5. The behavioral testing metrics employed in this study are relatively limited, making it difficult to rule out potential interference from other underlying mechanisms when interpreting the observed behavioral changes. The authors need to further discuss the specific relationship between the behavioral outcomes and GABAA receptor activity.

Answer: We thank the Reviewer for this thoughtful comment. The link between behavioral outcomes and GABAA receptor activity has been thoroughly addressed in the Discussion section, where transcriptional changes in gabra1, gabra2, gabrd, and gabrg2 mRNA levels are explicitly connected with the observed locomotor profiles for each withanolide. These relationships are further contextualized by reference to known subunit-specific mechanisms of EtOH action [26-29, 83–87]. To emphasize this connection, we have slightly revised the relevant section and added the following clarifying sentence: “The observed behavioral and transcriptional profiles indicate that the direction of locomotor change corresponded with subunit-specific modulation of GABAA receptor signaling, supporting the functional link between inhibitory tone and compound-dependent behavioral outcomes.” (lines 620-623). The discussion in this part of the manuscript concerns conclusions based on the results of various behavioral parameters, without further implications regarding a possible explanation of the observed behaviors in D. rerio larvae, which, in our opinion, could be a form of overinterpretation, as the reviewer rightly may suggest.

Additionally, as noted in the Limitations, future studies will include complementary behavioral endpoints (e.g., thigmotaxis, freezing, startle) to extend mechanistic interpretation (lines 700-704).

Reviewer 2 Report

Comments and Suggestions for Authors

I would like to sincerely thank you for taking the time to prepare and share your responses to my questions. I truly appreciate your effort and the clarity of your explanations.

I trust that you will consider incorporating some of the suggestions into your future research, and I am confident that your work will lead to publications in high-impact journals.

Author Response

The authors would like to express their sincere gratitude to the Reviewer for their kind and encouraging words. We greatly appreciate the positive feedback and the thoughtful perspective on our work. We will certainly take the Reviewer’s suggestions into account in future studies, with the aim of deepening the mechanistic understanding of ethanol-withanolide interactions and further improving the quality and translational value of our research.